# The Related Risk Factors of Diabetic Retinopathy in Elderly Patients with Type 2 Diabetes Mellitus: A Hospital-Based Cohort Study in Taiwan

**DOI:** 10.3390/ijerph18010307

**Published:** 2021-01-04

**Authors:** Tsai-Tung Chiu, Tien-Lung Tsai, Mei-Yin Su, Tsan Yang, Peng-Lin Tseng, Yau-Jiunn Lee, Chao-Hsien Lee

**Affiliations:** 1Department of Health Business Administration, Meiho University, Pingtung 912009, Taiwan; chiu12012001@gmail.com (T.-T.C.); x00002115@meiho.edu.tw (T.Y.); 2Lee’s Endocrinology Clinic, Pingtung 900028, Taiwan; lee@leesclinic.org; 3Department of Applied Mathematics, National Pingtung University, Pingtung 900391, Taiwan; tltsai@mail.nptu.edu.tw; 4Athletic Department, National Taiwan University, Taipei 106216, Taiwan; meiyin@ntu.edu.tw; 5Department of Nursing, Pingtung Christian Hospital, Pingtung 900026, Taiwan; yahapply@gmail.com; 6Department of Nursing, Meiho University, Pingtung 912009, Taiwan

**Keywords:** health behavior, chewing betel nut, dilated fundus examination

## Abstract

Diabetic retinopathy (DR), caused by small vessel disease, is the main cause of blindness in persons with diabetes. Taiwan is one of the Asian countries with the highest prevalence rate of DR. The purpose was to investigate the related risk factors of DR in elderly patients with type 2 diabetes mellitus (T2DM), in Lee’s Endocrinology Clinic. 792 T2DM patients over 60 years old were invited to have an outpatient visit at least every three months, and all of them were asked to undergo a standardized interview and collect their blood samples. Significant factors were being female (adjusted hazard ratio (HR): 1.287; 95% CI, 1.082–1.531), higher glycated hemoglobin (HbA1c) (HR: 1.067; 95% CI: 1.016–1.119), higher mean low density of lipoprotein cholesterol (LDL-c) (HR: 1.004; 95% CI: 1.001–1.006), and chewing betel nut (HR: 1.788; 95% CI: 1.362–2.347). This study showed that gender, the behavior of chewing betel nut, HbA1c, and LDL-c are important factors for the development of DR in elderly patients with T2DM. It is suggested that patients should control their HbA1c and LDL-c and quit chewing betel nut to prevent DR. This suggestion applies especially to female patients.

## 1. Background

Diabetes is a common chronic metabolic disease worldwide. Patients with diabetes need to strictly control and manage their lifestyle to maintain their blood index such as blood glucose, pressure, and lipids to prevent vascular neuropathy and diabetic neuropathy. Without proper management, it might accelerate disease deterioration into cerebrovascular and coronary artery diseases, and diabetic retinopathy (DR). In addition, the complications pose a significant threat to the worsening of symptoms [1,2]. In 2015, according to the World Health Organizations (WHO), the global prevalence of diabetes was 8.8%. The International Diabetes Federation (IDF) reported at least 425 million people with diabetes worldwide in 2017 and predicted that number to reach 629 million patients between 20 and 79 years old by 2045, which indicates that 1 in 11 adults might experience diabetes and 87–91% of the patient have type 2 diabetes mellitus (T2DM). In addition, five million people worldwide are expected to die from diabetes, implying one death every six seconds. In 2013, diabetes accounted for 14.5% of all causes of death in the world, and the number of deaths caused by diabetes was higher than that from infectious diseases [3]. According to the Taiwan survey between 2013 and 2015, the prevalence rate of diabetes was 11.8%, which was 3% higher than the global prevalence rate of diabetes. The survey also reported that 2.275 million people had been diagnosed with diabetes in Taiwan and that the number was increasing by the rate of 25,000 patients every year [4]. As a result, diabetes has been one of Taiwan’s top five leading causes of death since 1987 [5]. The complications of diabetes not only affect the health of people but also lead to medical burden. Therefore, it is very important to explore the risk factors of diabetic complications that have a significant impact on patients with diabetes.

In 2011, the proportion of diabetes patients over 20 years old was around 8.0%, and it was estimated that more than 1.4 million patients were diagnosed with diabetes in Taiwan. The National Health Insurance medical expenses for diabetes are around nearly USD 0.6 billion per year, accounting for about 4% of Taiwan’s National Health Insurance Fund. Various complications and diseased organs are consequences of diabetes, including kidney disease, heart disease, stroke, diabetic retinopathy, vascular disease, etc. For example, hemodialysis was about USD 1.1 billion per year, which is the highest among medical expenses associated with diabetes. Furthermore, the proportion of hemodialysis patients caused by diabetes had reached 40.0%, which increased the medical resources burden worldwide [6]. In 2010, the global healthcare expenditure on diabetes patients reached USD 376 billion per year, which accounted for 12.0% of the world’s total medical expenses [7]. The medical expenses were four times higher for diabetes patients with vascular complications than for those without vascular complications [8]. Therefore, diabetes complication imposes an increasing economic burden on national health care systems worldwide. With the increased prevalence of diabetes, the number of deaths attributable to diabetes and medical burden continue to rise. Therefore, it is vital to manage the health condition of patients with diabetes.

DR caused by small vessel disease was the main cause of blindness in persons with diabetes [9]. It is a significant cause of blindness for the elderly in both developed (United States, United Kingdom) and developing countries (Taiwan), accounted for 5% of the total blindness around the world [10,11,12]. According to the research, there are 2.6% of the 32.4 million blind people were related to DR in 2010. The visual impairment cases caused by DR had increased from 1.9 million to 3.1 million from 1990 to 2010. In 2015, 1.9% of DR led to blindness, and 10.2% were visually impaired [13,14].

The prevalence rate of DR was 23.0% in Asia-Pacific, 12.1% in Hong Kong, 25.4% in Singapore, and 15.8% in Korea [15]. Compared to other Asian countries, the prevalence rate of DR in Taiwan was much higher, at 31.1% [16]. Several studies have indicated that treatment methods would affect the risk of DR [12,17,18,19]. In addition, well-maintained blood glucose, blood pressure, and cholesterol levels could reduce the risk of DR, prevent vascular complications, and decelerate DR’s progress towards deterioration [20,21,22]. However, the prevalence rate of DR is still high in Asia due to the increasing incidence of diabetes every year. Although Taiwan is one of the Asian countries with the highest prevalence rate of DR, there is little research on the risk factor of DR for patients with T2DM in Taiwan. For example, betel nut is the fourth most common psychoactive substance, and the chewing behavior is a typical social interaction in Asia, particularly in the South Pacific islands, Southeast Asia, Papua New Guinea, Bangladesh, Pakistan, and India [23]. Betel nut chewing was linked to various health problems, including arterial stiffness, hypertension, obesity, diabetes mellitus, and periodontal disease [23,24,25,26,27]. Therefore, the main purpose of this study is to discover the related factors that affect the risk of DR in patients with T2DM. Furthermore, the study aims to improve the awareness and knowledge pertaining to the risk of DR in patients with T2DM as well as to reduce the incidence of DR among T2DM patients.

## 2. Methods

### 2.1. Subjects

From July 2016 to December 2017, in Lee’s Endocrinology Clinic, we recruited 792 patients who met these criteria and invited them for a standardized interview that included a dilated fundus examination. An ophthalmologist performed the dilated fundus examination (DFE) to evaluate and determine whether the patients with T2DM had progressed to DR. Inclusion criteria of study samples were as follows: (I) patients who were over 60 years of age and (II) patients who performed regular clinic visits for diabetes checkups and DFEs at least every three months during this study. Exclusion criteria of study samples were as follows: (I) patients who had monocular or bilateral blindness before or during the study, (II) patients who had glaucoma not caused by diabetic diseases, and (III) patients who were failing to follow up or unwilling to be involved in this study.

### 2.2. Collection of Participants’ Characteristics and Risk Factors Associated with DR

We asked participants to fill out a brief questionnaire at the scheduled outpatient visit. The brief questionnaire included questions on gender, education level, body mass index (BMI), diabetic duration, family history of diabetes, and health habits. We collected patents’ blood samples during the outpatient visit to check for glycated hemoglobin (HbA1c), fasting plasma glucose, triglyceride (TG), total cholesterol (CHOL-T), high density lipoprotein cholesterol (HDL-c), low density lipoprotein cholesterol (LDL-c), and the estimated glomerular filtration rate (eGFR). The education level of participants was classified as follows: no qualifications, elementary school, and junior high school or higher. BMI was measured as weight divided by height squared (kg/m^2^). T2DM duration in patients with DR was defined as the time (in years) from the onset time of diabetes to the time of first diagnosis at each stage of DR progression; diabetes duration of T2DM patients without DR was defined as the time (in years) from the onset time of diabetes to the time of the latest DFE. Health habits of participants included smoking, drinking, chewing betel nut, and regular exercise behavior. Smoking behavior was defined as smoking any type of cigarette at least once a day for a period of 6 months or longer. Drinking behavior was defined as drinking any type of alcoholic beverage at least three times a week for a period of 6 months or longer. The behavior of chewing betel nut was defined as chewing areca nut at least once a day for a period of 6 months or longer. Regular exercise behavior was defined as exercising regularly, at least three times a week, for a period of 6 months or longer. Blood pressure (systolic blood pressure and diastolic blood pressure) was measured with a digital automatic blood pressure monitor with the participant seated after resting for 5 min. The levels of creatinine, fasting plasma glucose, TG, CHOL-T, and HDL-c were measured on an analyzer with an enzymatic assay. HbA1c was measured on an analyzer using high-performance liquid chromatography. The eGFR was estimated in accordance with the formula (186 × Creatinine^(−1.154) × Age^(−0.203) (×0.742 if female)) [27]. Moreover, LDL-c was calculated with the Friedewald formula, including TG, CHOL-T, and HDL-c [28].

### 2.3. DFE Procedure

To obtain a better view of the fundus of the eye, we performed DFE on all participants. The procedure included an eyesight test, automatic optometric examination, and air puff test. It is a diagnostic procedure that employs the use of mydriatic eye drops (such as Mydrin-P) to dilate or enlarge the pupil. The value of intraocular pressure must be reported as normal before the dilated eye exam. The ophthalmologist used the mydriatic eye drops, delivering the eye drop every 5 min thrice. After 30 min, the patient’s pupil would stay open with its size naturally enlarged, enabling the ophthalmologist to use ophthalmoscopy to view the eye’s interior, allowing for assessment of the retina, blood vessels, optic nerve head, and other features. It also allowed the ophthalmologists to diagnose and monitor the degree of DR in a more detailed manner once the digital images from all participants were obtained.

### 2.4. Statistical Analysis

The participants’ characteristics were described by absolute and relative frequency for categorical variables and mean and standard deviation for continuous variables. In inference statistic methods, an independent sample T-test was used to analyze the difference of the mean for continuous data between the patients with and without DR. A chi-square test was used to analyze the difference in the proportion of categorical variables between the patients with and without DR. Pearson product–moment correlation coefficients were applied to test the correlations between each risk factor. Finally, the significant factors in each statistic testing were considered in the model. The optimal multivariate Cox regression model was determined by stepwise model selection. Those results were reported as hazard ratios (HR) with a 95% confidence interval (CI). A *p*-value < 0.05 was considered to be statistically significant. Statistical analysis was performed using IBM SPSS Statistics 24.

## 3. Results

In this study, 792 patients with T2DM were effectively enrolled. Their average age was 67.85 (SD = 6.53). During the follow-up period, DR was diagnosed in 611 patients (77.15%). In this study, there were 339 male patients and 453 females. About 493 patients reported a history of diabetes in their family, 149 patients reported drinking behavior, and 79 patients were addicted to chewing betel nut. A family history of diabetes was reported by 493 patients, 149 patients reported drinking behavior, 79 patients were addicted to chewing betel nut, 108 patients reported no educational qualifications, 189 patients were addicted to smoking behavior, and 453 patients reported regular exercise behavior.

According to the results in Table 1, the significant predictive factors for DR in elderly patients with T2DM included gender (χ^2^ = 8.99, *p* = 0.003), a family history of diabetes (χ^2^ = 11.79, *p* = 0.001), and a behavior of chewing betel nut (χ^2^ = 3.97, *p* = 0.046). However, there was no significant association between education level, smoking behavior, drinking behavior, or regular exercise behavior and the development of DR.

In Table 2, the factors HbA1c (t = −4.61; *p* < 0.001), TG (t = −2.39; *p* = 0.017), CHOL-T (t = −4.44; *p* < 0.001), and LDL-c (t = −3.26; *p* = 0.001) were associated with the development of DR in elderly patients with T2DM. Moreover, we also found that HbA1c had a significant positive correlation with TG, CHOL-T, and LDL-c, and the correlation coefficients were 0.147, 0.248, and 0.220, respectively. The correlation coefficient for TG versus CHOL-T was 0.337, for TG versus LDL-c was 0.193, and for CHOL-T and LDL-c was 0.845; the *p*-value of each group was under 0.001 significant rate. However, factors such as BMI, systolic blood pressure, diastolic blood pressure, fasting plasma glucose, HDL-c, and eGFR were not significantly associated with the development of DR in elderly patients with T2DM.

Integrating the results from Table 1 and Table 2, factors including gender, behavior of chewing betel nut, family history of diabetes, HbA1c, TG, CHOL-T, and LDL-c were associated with the development of DR. With Cox regression analysis by stepwise model selection, we found that the significant factors associated with the occurrence of DR in elderly patients with T2DM, retained in the optima model, were being female (HR: 1.287; 95% CI, 1.082–1.531), higher HbA1c (HR: 1.067; 95% CI: 1.016–1.119), higher LDL-c (HR: 1.004; 95% CI: 1.001–1.006), and the behavior of chewing betel nut (HR: 1.788; 95% CI: 1.362–2.347), as showed in Table 3.

## 4. Discussion

This study found that factors such as gender, the behavior of chewing betel nut, HbA1c, and LDL-c were important predictors of DR in elderly patients with T2DM. The risk of DR in female patients was higher than that in male patients. The risk of retinopathy in patients with a behavior of chewing betel nut was higher than that in patients without a behavior of chewing betel nut. Patients with higher HbA1c and higher LDL-c had a higher risk of developing retinopathy. Most of our findings were consistent with risk factors reported in existing literature; additionally, we observed the behavior of chewing betel nut to be a unique risk factor to the Taiwanese population.

Regarding gender, other studies previously reported that women are more likely to develop retinopathy than men [29,30]. Estrogen production may regulate ocular blood flow to protect the retina; its antioxidant effects are the primary protective effect on the lens. When women age beyond 50, the concentration of estrogen decreases year by year and gradually its protective function is lost [31,32]. The average age of women in our study was over 65 years. This may be why the risk of DR was found to be higher in females.

The results of a recent study of relationships between diabetes duration and risk of DR in patients with T2DM showed that the prevalence of retinopathy was 1.1% at the first time of diagnosis, 6.6% for diabetes duration of less than 5 years, 12.0% for diabetes duration between 5 and 10 years, 24.0% for diabetes duration between 10 and 15 years, 39.9% for diabetes duration between 15 and 20 years, and 52.7% for diabetes duration over 20 years. Therefore, the prevalence of DR in patients rises substantially when their diabetes duration is over 10 years [33]. Some studies also found that the longer the duration of diabetes, the higher the risk of DR development [17,30,34]. The result of these studies corresponds to our study. Therefore, early control can prevent and delay the progression of DR in patients with diabetes.

Lopez et al. (2017) found that patients with a family history of diabetes had a higher chance of developing DR; the number of patients with a family history of diabetes was about 63.4% [29]. Genetic inheritance and lifestyle were found to have a great influence on the induction of T2DM [35]. Therefore, T2DM patients with a family history of diabetes should definitely manage their lifestyle and control diabetes. Keeping the blood glucose level balanced is one of the most important steps that one should take. A consistently high blood glucose level gradually leads to organ damage in the long term [36].

In terms of blood biochemical values pertaining to blood pressure and BMI, the results of this study showed that HbA1c, TG, CHOL-T, and LDL-c were highly associated with the risk of DR in elderly patients with T2DM. On the other hand, systolic blood pressure, diastolic blood pressure, fasting plasma glucose, HDL-c, eGFR, and BMI were not significantly associated with the risk of DR. A study explained that higher HbA1c increases the risk of DR and that hyperglycemia causes injury to body organs, such as nerve damage (neuropathy), kidney damage (diabetic nephropathy), and even damage to the blood vessels of the retina (diabetic retinopathy), potentially leading to vision loss [37]. Stratton et al. (2000) pointed out that every 1% reduction of HbA1c can decrease the incidence of cerebrovascular accidents by 12%, heart failure by 16%, and amputation or death caused by peripheral arterial occlusive disease by 43% and reduce small blood vessel diseases by 37% [38]. One study in China also found that higher systolic blood pressure, HbA1c, fasting plasma glucose, and LDL-c and lower TG are important factors that increase the risk of DR [39]. However, in another study in China, higher systolic blood pressure and HbA1c and lower BMI were associated with the presence of DR [40]. A study in South Korea also confirmed that higher HbA1c is a significant risk factor for DR in patients with T2DM [34]. The increased risk of DR was found to be caused by higher HbA1c and higher LDL-c in the above studies, which is consistent with our study, but TG and CHOL-T were not. Thus, control and management of HbA1c and LDL-c should significantly decrease the chance of developing DR or prevent the progression of diabetic retinopathy in patients with diabetes [1,14,18,41,42,43].

Moreover, this study found that the behavior of chewing betel nut, which is a common social interaction throughout Asia, is significantly associated with the development of DR in elderly patients with T2DM and is an essential predictor of DR. Some studies found that the behavior of chewing betel nut was associated with newly diagnosed T2DM and the occurrence of chronic kidney disease and increases the risk of cardiovascular disease [23,25,26]. Others found that chewing betel nut more frequently increased blood pressure. The behavior of chewing betel nut was also associated with an increased risk of arterial stiffness [24]. Therefore, it is possible to suggest that the behavior of chewing betel nut can affect small vessel disease and increase the risk of DR. Its biological mechanism might be a possible linkage between betel nut chewing and the development of microvascular complications that necessitates further discussion.

This study found that the characteristics in the high-risk group of developing DR were being female, a family history of diabetes, and the behavior of chewing betel nut. Therefore, it is suggested that patients with these factors should always keep their HbA1c, LDL-c, TG, and CHOL-T in a normal range and quit their behavior of chewing betel nut to prevent developing DR. Moreover, integrated care focusing on more coordinated and integrated forms of care provision may play an important role in the care of patients with diabetes mellitus. Not only do blood glucose, blood pressure, and blood lipids have to be regularly controlled and managed for patient with diabetes, but related complications of diabetes should also be regularly screened. However, anyone who has diabetes mellitus is always under the risk of developing DR; some may be diagnosed with DR after having diabetes mellitus for several years. Thus, it is important to enable clinical care personnel to promote regular health examinations for T2DM patients by exploring and understanding the important predictors of DR in patients with T2DM. Early intervention for vision problems is useful to prevent severe vision loss. Doctors should also allocate patients to appropriate hospitals or clinics in time for patients to attend to routine pupil examinations and consultations. Nurses should also implement the education of health guidelines individually. It will be helpful to improve the self-management abilities of patients who are under a high risk of the progression of DR.

## 5. Conclusions

This research was a hospital-based study, and the patient recruitment might not represent overall diabetes patients. We cannot exclude the residual confounding factors due to unmeasured factors. To better understand the relationship between DR and relative risk factors, it is necessary to have a more extensive population study. Our study showed that gender, the behavior of chewing betel nut, HbA1c, and LDL-c are important factors for the development of DR in elderly patients with T2DM. We suggested that patients should control their HbA1c and LDL-c and quit chewing betel nut to prevent DR, especially female patients. However, our study presents that the behavior of chewing betel nut is one of the potential risk factors for DR in elderly patients with T2DM, a result that is very different from the other studies. Therefore, the relationship between chewing betel nut and the risk of DR in diabetic patients needs further investigation.

## Figures and Tables

**Table 1 ijerph-18-00307-t001:** Comparison of categorical variables between the patients with and without DR (N = 792).

Variables		No DR (N = 181)		DR (N = 611)		χ^2^	*p*-Value
		n	%	n	%		
Gender	Male	95	28.02	244	71.98	8.99 **	0.003
	Female	86	18.98	367	81.02		
Educational qualifications	No qualifications	22	20.37	86	79.63	1.17	0.557
	Elementary school	66	21.71	238	78.29		
	Junior high school or higher	93	24.47	287	75.53		
Family history of diabetes	No	88	29.43	211	70.57	11.79 **	0.001
	Yes	93	18.86	400	81.14		
Smoking behavior	No	134	22.22	469	77.78	0.57	0.450
	Yes	47	24.87	142	75.13		
Drinking behavior	No	145	22.55	498	77.45	0.18	0.673
	Yes	36	24.16	113	75.84		
Chewing betel nut behavior	No	170	23.84	543	76.16	3.97 *	0.046
	Yes	11	13.92	68	86.08		
Regular exercise behavior	No	73	21.53	266	78.47	0.59	0.444
	Yes	108	23.84	345	76.16		

Note: DR, diabetic retinopathy; * *p*-value < 0.05; ** *p*-value < 0.01.

**Table 2 ijerph-18-00307-t002:** Comparison of continuous variables between the patients with and without DR (N = 792).

Variables	No DR (N = 181)	DR (N = 611)	t	*p*-Value
	Mean ± SD	Mean ± SD		
Body mass index (kg/m^2^)	26.09 ± 4.06	26.27 ± 4.12	−0.51	0.610
Systolic blood pressure (mmHg)	136.48 ± 19.10	139.13 ± 20.69	−1.55	0.123
Diastolic blood pressure (mmHg)	76.27 ± 11.67	77.73 ± 12.51	−1.40	0.161
HbA1c (%)	7.66 ± 1.47	8.25 ± 1.68	−4.61 ***	<0.001
Fasting plasma glucose (mmol/L)	8.31 ± 2.94	8.76 ± 3.33	−1.61	0.108
TG (mmol/L)	1.45 ± 0.64	1.59 ± 0.87	−2.39 *	0.017
CHOL-T (mmol/L)	2.00 ± 0.33	2.13 ± 0.43	−4.44 ***	<0.001
HDL-c (mmol/L)	0.58 ± 0.16	0.58 ± 0.15	−0.27	0.790
LDL-c (mmol/L)	1.04 ± 0.27	1.12 ± 0.34	−3.26 **	0.001
eGFR (mL/min/1.73 m^2^)	72.18 ± 21.20	70.21 ± 27.08	0.89	0.377

Note: DR, diabetic retinopathy; HbA1c, glycated hemoglobin; TG, triglycerides; CHOL-T, total cholesterol; HDL-c, high density lipoprotein cholesterol; LDL-c, low density lipoprotein cholesterol; eGFR, estimated glomerular filtration rate; * *p*-value < 0.05; ** *p*-value < 0.01; *** *p*-value < 0.001.

**Table 3 ijerph-18-00307-t003:** A Cox proportional hazard model for risk factors of DR in patients with T2DM (N = 792).

Variables	ß	SE	Wald	HR	95% CI		*p*-Value
HbA1c (%)	0.064	0.025	6.851	1.067 **	1.016	1.119	0.009
LDL-c (mg/dL)	0.004	0.001	8.455	1.004 **	1.001	1.006	0.004
Gender	Reference						
Male							
Female	0.252	0.089	8.136	1.287 **	1.082	1.531	0.004
Chewing betel nut behavior	Reference						
No							
Yes	0.581	0.139	17.543	1.788 ***	1.362	2.347	<0.001

Note: DR, diabetic retinopathy; T2DM, type 2 diabetes mellitus; HR, adjusted hazard ratio; CI, confidence interval; HbA1c, glycated hemoglobin; LDL-c, low density lipoprotein cholesterol; ** *p*-value < 0.01; *** *p*-value < 0.001.

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
