# Peer review of "The Related Risk Factors of Diabetic Retinopathy in Elderly Patients with Type 2 Diabetes Mellitus: A Hospital-Based Cohort Study in Taiwan"

_ijerph, 2021, doi:10.3390/ijerph18010307_

Round 1
Reviewer 1 Report
Major comments:
» The paper is not written in a very fluent English and sometimes makes it difficult to perceive the idea that it is intended to be conveyed
» lines 74-75: the authors say “DR caused by small vessel disease was the main cause of blindness in persons with diabetes and mainly occurs in patients with T2DM” and cite a paper with more than 20 years. To my knowledge, to the present day there is evidence that diabetic retinopathy is more prevalent in type 2 diabetes compared with type 1 diabetes patients; in absolute numbers, the majority of diabetic retinopathy cases are among patients with type 2 diabetes because it’s the most common form of diabetes. I suggest that the sentence should be replaced by “DR caused by small vessel disease was the main cause of blindness in persons with diabetes”. Also in the abstract (lines 17-18) the same applies: instead of “Diabetic retinopathy (DR), caused by small vessel disease, is the main cause of blindness in persons with diabetes and mainly occurs in patients with Type 2 diabetes mellitus (T2DM)” should be “Diabetic retinopathy (DR), caused by small vessel disease, is the main cause of blindness in persons with diabetes”.
» lines 120-123: the authors say “Diabetic duration of T2DM patients with DR was defined as the time (in years) from the onset time of diabetes to the time of first diagnosis at each stage of DR progression; diabetic duration of T2DM patients without DR was defined as the time (in years) was from the onset time of diabetes to the time of the latest DFE”. As we don’t know when T2DM develops but the date when the disease is diagnosed, I suggest that the sentence should be “T2DM duration in patients with DR was defined as the time (in years) from the onset time of diabetes to the time of first diagnosis at each stage of DR progression; diabetic duration of T2DM patients without DR was defined as the time (in years) was from the onset time of diabetes to the time of the latest DFE”
» lines 134-135: The authors say “The eGFR was measured with the level of creatinine in the blood, using the result in a formula to calculate a number that reflected how well the kidneys were functioning.” It should be stated which formula was used and to be included the bibliographic reference concerning that formula. Moreover, the last part of the sentence is not very scientific and should be abolished. It should be substituted by the sentence “The eGFR was estimated in accordance to the formula XXXXX.”
» lines 136-137: The authors say “Moreover, LDL-c was calculated with a formula including TG, CHOL-T, and HDL-c.” It should be “Moreover, LDL-c was calculated with the Friedewald formula.” placing the appropriate bibliographic reference.
» lines 163-164: It is written that “During the follow-up period, 611 patients (77.15%) progressed to DR but 181 patients (22.85%) did not.” However, the sentence was true if no one presented with DR at baseline and, during the 18 months period, they have developed that complication. So, it should be replaced by “DR was diagnosed in 611 patients (77.15%).”
» lines 164-166: It is written “About 493 patients reported a history of diabetes in their family, 149 patients reported drinking behavior, and 79 patients were addicted to chewing betel nut.” The word “about” is used when we do not say the exact number and that is not the case. If authors mentioned the prevalence of these patients’ characteristics, they also should be mentioning the prevalence of other characteristics analyzed. So, the sentence should be “A family history of diabetes was reported by 493 patients, 149 patients reported drinking behavior, 79 patients were addicted to chewing betel nut, XXX educational qualifications, XXX smoking behavior and XXX regular exercise behavior.”
» lines 207-210: The authors say “Regarding the gender factor, Lopez et al. (2017) determined that retinopathy in women are more common compared to retinopathy in elderly patients with T2DM.23 Another retrospective study also found that women are more likely to develop retinopathy than men24, the results of which matched the results of this study.” The sentence is too confusing and should be simplified like “Regarding gender, other studies have previously reported that women are more likely to develop retinopathy than men23,24.”
» lines 210-214: The authors describe a hypothetical explanation for the higher prevalence of DR among women stating “The estrogen production may regulate ocular blood flow to protect the retina; its antioxidant effects are the primary protective effect on the lens. When women age beyond 50, the concentration of estrogen would decrease year by year and, gradually, its protective function would be lost.25–26 The average age of women in our study was over 65 years. These may be the reasons explaining why the risk of DR was found to be higher in females.” However, the question is not to find an explanation for the difference between pre and post-menopausal women but instead a difference between men and women. Could the difference be attributable to a higher prevalence of other risk factors for DR among women, namely chewing betel nut? The explanation should be revised.
» lines 220-222: Instead of the sentence “Some studies also found that the longer the duration of diabetes, the higher is the chance of retinopathy appearing, impacting the risk of DR”, which is too redundant, it is better to write “Some studies also found that the longer the duration of diabetes, the higher the risk of DR development”
» Conclusions: The authors have mentioned at the conclusion item of the abstract that “This study showed that gender, behavior of chewing betel nut, HbA1c, and LDL-c are important factors for the development of DR in elderly patients with T2DM. It suggested that patients should control their HbA1c and LDL-c and quit chewing betel nut to prevent DR. This suggestion applies especially to female patients.” However, in the conclusion section of the paper they do not make any reference to those risk factors, with the exception for chewing betel nut. So, this section should be revised.
Minor comments:
» throughout the paper, text and tables, it is written “Family history of diabetic” and “diabetic duration” instead of “Family history of diabetes” and “diabetes duration”, respectively
» line 31: instead of “…higher concentration of glycated hemoglobin (HbA1c)…” it’s better “…higher glycated hemoglobin (HbA1c)…”
» line 113: Instead of “We asked participants to fill out brief questionnaire at the scheduled outpatient visit” it should be “We asked participants to fill out a brief questionnaire at the scheduled outpatient visit”
» line 139: Instead of “To obtain a better view of the fundus of the eye, we performed DEF on all participants.” it should be “To obtain a better view of the fundus of the eye, we performed DFE on all participants.”
» line 184: Instead of “HbA1c, glycosylated hemoglobin” it should be “HbA1c, glycated hemoglobin”
» lines 220-222: Instead of the sentence “Some studies also found that the longer the duration of diabetes, the higher is the chance of retinopathy appearing, impacting the risk of DR”, which is too redundant, it is better to write “Some studies also found that the longer the duration of diabetes, the higher the risk of DR development”
» lines 258-259: The authors write that “This study found that the demographic characteristics in the high-risk group of developing DR are being female, family history of diabetes, and behavior of chewing betel nut.” However, chewing betel nut is not a demographic characteristic. It should say “This study found that the characteristics in the high-risk group of developing DR are being female, family history of diabetes, and behavior of chewing betel nut.”
Reviewer 2 Report
In the introduction and/or discussion I would recommend the authors to develop the main finding here, i.e the "behavior of chewing betel nut" and DR.
For example, authors may remind the nature and the cultural habit of betel nut chewing (Indeed, Betel nut is the fourth most commonly used psychoactive substance, in Asia, particularly the South Pacific islands, Southeast Asia, India...
In the discussion: physiopathological hypothesis may be cited and discussed in order to understand the possible link between betel nut chewing and the development of microvascualr complications. Indeed, Betel nut chewing has been linked to a variety of health problems including arterial stiffness, hypertension, obesity, diabetes mellitus and periodontal disease. (See ref : https://doi.org/10.1186/1756-0500-3-228 and https://doi.org/10.1016/j.drugalcdep.2017.07.035)
Author Response
Please see the attachment.

This manuscript is a resubmission of an earlier submission. The following is a list of the peer review reports and author responses from that submission.
Round 1
Reviewer 1 Report
This is a 500 day epidemiologic study conducted by a business professor, an athletics professor, two nurses, and an endocrinologist re-demonstrating certain known characteristics regarding the risk of developing diabetic retinopathy in a patient population most known to be at risk of the cormorbidity (a1c, hyperlipidemia, age, etc). Authors introduce one variable aspect more common to the Southeastern Asian population: namely the consumption of the Areca or betel nut. Their description of process is lacking, they only describe the minimum inclusion criteria and the final results of their study; importantly they do not describe any baseline characteristics nor do they show when diabetic retinopathy was diagnosed in the followup period. Furthermore, they do not describe which examination findings supporting their theory as to the presence of diabetic retinopathy: microaneurysms, intraretinal hemorrhages, dot-blot hemorrhages, cotton wool spots, intraretinal microvascular anomalies (IRMA), neovascularization of the disc or elsewhere, vascular beading, or sclerosis. While seemingly unimportant, some of these findings are present in hypertensive retinopathy and they show in their results that this patient population has an average systolic blood pressure above 130 mm Hg. Presumably much, some, or all of the findings supporting a diagnosis of DR could be attributed to hypertensive retinopathy.
Gender seems to be an area of focus in this piece however I see no direct comparisons within the Tables or paragraph descriptions presented. Please define this more clearly. What was the odds ratio for DR in males compared with females? Why are males considered the reference?
Which photography technique was employed? In most reliable epidemiologic studies a 7 field 50 degree camera is used as was first described in Wisconsin, USA in the 1980s. Did the authors' reference ophthalmologist use a single 50 degree image? Did they use 7 fields? Did they use widefield single image? In the Joslin studies a 2 field, widefield camera was used. These days most clinical trials still use 7 field though some have began to use ultrawidefield (Optos, for example). Examination of the retina is a challenge and photography is a necessary ancillary test for thorough analysis, monitoring step progression, etc. Inclusion of the subspecialist as an author may clear this confusion in addition to exam findings typical of DR described above.
This publication contradicts many important, larger, longer longitudinal studies when they show that those with T2DM duration around 15 years do not have diabetic retinopathy. Duration of diagnosis is known to be the most consistent relationship. In WESDR 50% of men and 30% of women had proliferative diabetic retinopathy (end stage, highest grade) within 19-20 years. Harris (Diabetes Care 1992;15:815-9, a researcher not cited in this work, extrapolated data from WESDR and Blue Mountain study) estimated that onset from diagnosis is approximately 4-7 years. Why does this publication contradict these findings in your view?
I would suggest, as a primer, that the authors study the history of diabetic retinopathy as it is a large subject requiring many references to a literature that started in the 1980s in Wisconsin until the present day DRCR, Proyecto, etc. One useful resource is Ryan's retina text, chapter 45, or any review written by Barbara Klein.
The English language needs a lot of work in this publication. I would suggest the authors consult with a native speaker.
Reviewer 2 Report
Here, Chiu et al. presents the evaluation of selected clinical and biochemical variable in a cohort of 792 patients with type 2 diabetes. Patients were evaluated for the development of diabetic retinopathy over a 18-month follow up period.
The major of this study is the unclarity of the study design. The study is designed as a cohort study, with a single group of patients being monitored over a predefined time frame. However, results are instead presented as a case-control studies, with redundant logistic regression analyses for each variable followed by an overall logistic regression encompassing all the variables. Given the study design, I would have presented a Cox proportional hazard model rather than a logistic regression.
Moreover, the variables associated (or predictive?) with DR do offer none or incremental advance over what is already known on the factors associated with the onset of T2DM complications (i.e. smoking, disease duration, HbA1c, lipid profile)
For these reasons, introduction and discussion are way too longer than needed, as well as the results, which report information that is already listed in the tables.
Reviewer 3 Report
This paper examines the risk factor associated with diabetic retinopathy using cross-sectional data collected between 2016 to 2017 from 792 T2DM patients over 60 years old residing in Taiwan. The study showed that gender, behavior of chewing betel nut, diabetic duration, family history of diabetes, HbA1c, and LDL-c are important factors for the development of DR in elderly patients with T2DM. Overall, the paper is well written; it had clear literature review and description of the method and the results. The study design and analytical methods were appropriate, and the finding do provide value to the clinical practice of this field, especially for treating older T2DM patients in Taiwan. I would recommend the journal accept the paper for publication with revisions.
Here are my major comments:
- Please use active voice rather passive voice. For example, rather than saying the study were conducted”, say “we conducted the study”
- Please add a correlation test on the risk factors; it will be interesting to see the relationship between each risk factors.
- The paper can be strengthened by adding more description of the study population and assessment of any potential for selection bias. This will also help address the external validity (generalizability) of the finding.
- Please add a paragraph on the strength and limitations of the study in the discussion. Specifically, please assess internal and external validity of the results.
Other minor comments:
Line 48: please provide more information on the prevalence of diabetes (was 8.8% the global prevalence?)
Line 51: change the sentence to future tense “This means that 1 in 11 adults will experience diabetes with 87-91% having type 2 diabetes mellitus.
Line 62: citation for the 8% prevalence. Does 8% reflects 1.4 million patients? I suggest combining the two sentences to make it clearer. The way it is written now made the two sentences seems like they’re conveying different messages.
Line 64: add the equivalent of 18.4 billion Taiwan dollar to USD or Euro so it’s easier for the global audience
Line 65: was the 4% annual expenditure for the entire national expenditure in Taiwan or the annual expenditure for the National Health Insurance?
Line 77: With the “increased” or “high” prevalence of diabetes?
Line 83-87: please provide the region/country associated with each statistic and try to report numbers rounding to the same unit (e.g. just one unit below the decimal point)
Line 90: change the word “reached 31.1%” to “was much higher at 31.1%”
Line 98: change “consciousness” to “awareness”
Line 100: change “to the reduction of the incidence rates of DR in patients with T2DM” to reduce the incidence of DR among T2DM patients”
Line 103: Please try to use active voice throughout the paper, for example rather than saying patients were in accordance with the study sample inclusion criteria, say, we set the following inclusion and exclusion criteria, which were….” “We recruited 792 patients who met these criteria and invited them for a standardized interview which included a dilated fundus examination….”.
Line 113: rather than “materials and methods”, change the subtitle of this section to “collection of participants characteristics and risk factors associated with DR”, and then use active voice to describe the collection of data. For example, you can start the paragraph with “We asked participants’ to fill out brief questionnaire at the scheduled outpatient visit. The brief questionnaire included questions on gender, education level……” “We collected patents’ blood sample during the outpatient visit (four to five times a year?) to check for glycated hemoglobin, fasting plasma glucose,……”
Line 122: how was diabetic during and health habits collected? Interview by research staff?
Line 135: please cite the formula used for the eGFR estimation since there are several that are regularly used in clinical settings. Did you use the one with serum creatinine and serum cystatin? Was this formula designed for the Taiwanese population?
Line 140: again, please use active voice here. For example, you can change the sentence to “to obtain a better view of the fundus of the eye, we performed DEF on all participants. The procedure included eyesight test, automatic optometric examination, and air puff test”.
Line 159: did you check for correlation between the variables included in the logistic regression? It will be interesting to show their correlation structure.
Line 167: instead of using “were with”, I suggest changing it to “reported”. For example, 149 patients reported drinking behavior”
Line 164-168: you can just report those with the behavior. No need to report % without since they added up to 100%
Table 1: “Male” was bolded which might be a word-processing error, There is no HR in the table (but it was mentioned in the note), in the note, please also specify whether the OR is derived from univariate logistic regression (crude OR?), and note what * and ** stands for
Table 2: again, diabetes duration was bolded, likely due to processing errors. HR was not mentioned in the table and please note what *, **, *** stands for
Line 224: it would be nice if you provide a high-level summary of which finding was consistent with existing literature and which findings were novel. You can say something like, “most of our finding were consistent with risk factors reported in existing literature, additionally, we observed xxx risk factors to be a unique risk factor to the Taiwanese population (eg. betal nut chewing)
Line 225: What was the study setting and participants’ characteristics of Lopez’s study? Please elaborate more on that and how do those factors affect the comparison of results.
Round 2
Reviewer 1 Report
In my first review I had offered the authors advice, I specified that inclusion of an ophthalmologist into the authors list would be helpful. One reason for this is because they are very light in detail on the description of diabetic retinopathy (how it was diagnosed, what criteria were used, which camera is used, etc) and could not provide justification for specific confounders in the diagnosis of diabetic retinopathy. For example: it is very common for patients with diabetic retinopathy to demonstrate intraretinal hemorrhages but most importantly predominant microaneurysmal vasculopathy. This publication does not list one physical exam finding/criteria supporting the diagnosis of DR. I provided a potential confounder, namely hypertensive retinopathy, that must be accounted for as those with hypertension can also demonstrate intraretinal hemorrhages without a predominant microaneurysmal vasculopathy. I'm not saying exclude all hypertensives, I'm saying the character of the retina determines whether or not the retinal finding most suggests diabetic retinopathy or hypertensive retinopathy. If they are unaware of these subtle but very important differences, they would find new changes within the retina that would be documented in their case incidence of diabetic retinopathy when in fact their patients could merely have microvascular changes merely consistent with hypertensive retinopathy. This is called counfounding in clinical research and it is very important. You may find an association of this or that behavior, use of food or medication and document a rise in exam findings that may actually support a completely different theory when the variable is not accounted for. In their response they show a lack of understanding. For example:
"Although high blood pressure might be the leading risk factor to induce DR. In our study, most of our patient’s blood pressure was under, which did not reach the baseline of high blood pressure."
In this they demonstrate a lack of understanding of either statistics (namely that if the average systolic BP is ~130, that two standard deviations in a Gaussian distribution will have a systolic over 140) or of the nature of blood vessel changes that always occur in the retina in any patient who has a history of hypertension regardless of whether or not there is either a difference between DR and no DR groups or whether or not the systolic and diastolic blood pressures are recorded below 140 or 90 in their clinics. In fact they add a new table, Table 2, which shows an average systolic pressure of 136 +/- 19 in the no DR group and 139 +/- 20! My friends, this is not normal blood pressure. This demonstrates to me that they did not adequately conceive of potential reasons for findings that are potential confounders for a finding within the retina that would be documented as having DR.
To correct this I suggest the authors contact their ophthalmologist and ask him or her the following: what specific exam features did you use to identify patients as having diabetic retinopathy? Did you diagnose diabetic retinopathy when you found microaneurysms? intraretinal hemorrhages? cotton wool spots? intraretinal microvascular anomalies (IRMA)? neovascularization? Did you diagnose diabetic retinopathy only when you found a combination of all of them? Did you diagnose diabetic retinopathy on the basis of your hunch? Was there a microaneursymal predominance? Or did you use criteria such as merely intraretinal hemorrhages that would be found in a population with a history of hypertension (as this population clearly demonstrates in your presented findings)?
The authors provided some details regarding the camera used but do not list this camera in the manuscript. They inform me that they used a Topcon SL-D7. Dear authors, this is not a camera. The "SL" in Topcon SL-D7 stands for "slit lamp." A slit lamp is a tool used to examine patients no different than a stethoscope or a tongue depressor. A camera is a separate device that captures an image of the retina while light is simultaneously flashed at the retina. The resulting image is viewable on a computer or paper printout. There are many camera manufacturers, Topcon makes at least 6 cameras for example: TRC-NW8F, TRC-NW4000, TRC-NW8, TRC-50DX, TRC-NW8F, and NW7SF. In the TRC nomenclature they are highlighting the most important aspect of photography: Topcon Retinal Camera (TRC)... that this device is, in fact, a camera, not a slit lamp (SL). Furthermore, the documentation of which kind of photography technique would be helpful and is not addressed in the reply or manuscript. One could take a single 50 degree image centered on the macula and not capture 90% of the retinal surface, thereby missing diabetic retinopathy, hypertensive retinopathy, etc one would otherwise document in a study such as this. Knowing the photography technique may actually provider *greater* evidence for the variables they are claiming produce diabetic retinopathy if they used a photography methodology that can capture the appearance of the retina adequately. Was a image composite created? if so, what kind of montage? 7 image (as was used in WESDR)? 2 wide-field images (as was used in Joslin)? What non-ophthalmologists do not realize is that examination of the retina is a very difficult thing to do and photography is an excellent way of taking a detailed examination without the challenge of shining very bright light sustained at a very sensitive tissue. Again, inclusion of an ophthalmologist in this study would clarify this point.